# Relationships of Social Support, Stress, and Health among Immigrant Chinese Women in Japan: A Cross-Sectional Study Using Structural Equation Modeling

**DOI:** 10.3390/healthcare9030258

**Published:** 2021-03-01

**Authors:** Yunjie Luo, Yoko Sato

**Affiliations:** 1Graduate School of Health Sciences, Hokkaido University, Sapporo, Hokkaido 060-0812, Japan; raa@eis.hokudai.ac.jp; 2Faculty of Health Sciences, Hokkaido University, Sapporo, Hokkaido 060-0812, Japan

**Keywords:** social support, parenting stress, acculturative stress, mental health, physical health, immigrants, women

## Abstract

Social support could help immigrant Chinese women in Japan to improve health. However, these women suffer from numerous stresses associated with acculturation and child-rearing, which could impact the effect of social support on mental and physical health. This study aims to identify the relationships among social support, acculturative stress, parenting stress, mental health, and physical health to propose a structural path model of these relationships and to evaluate the mediating role of stress between social support and health. A questionnaire was used to collect data for the main variables and demographic factors. A total of 109 women participated (mean age 33.9 ± 5.6 years). The results show that immigrant Chinese women in Japan experienced a low level of mental health (44.7 ± 9.8). Additionally, social support directly influenced parenting stress (*β* = −0.21, *p* < 0.05), acculturative stress (*β* = −0.19, *p* < 0.05), and mental health (*β* = 0.31, *p* < 0.001) and indirectly influenced physical health (*β* = 0.09, *p* < 0.05). Parenting stress partially mediated the association between social support and mental health (*β* = 0.09, *p* < 0.05). To protect the effectiveness of social support on mental health, it is essential to reduce the parenting stress of immigrant Chinese women.

## 1. Introduction

As of June 2019, the number of foreign residents in Japan exceeded 2,829,000, hitting a record high [1]. With the increasing number of foreign residents, the Japanese Ministry of Internal Affairs and Communications developed a Multicultural Coexistence Promotion Plan to help immigrants adapt to life in Japan [2]. This has been in place since 2006. Based on this plan, each region in Japan provides livelihood support for foreign residents, such as creating a maternal and child health handbook in multiple languages, expanding parenting classes, and providing interpretation services for parents in nursery schools. This is helping Japan to become a multicultural society. Among the foreign residents in Japan, immigrant women raising children constitute a noticeable proportion [3]. Despite having been provided with social support, these immigrant women still experience numerous difficulties associated with acculturation and child-rearing, such as adapting to a new culture, the language barrier, and mental and physical distress [3,4,5].

According to the Japanese Ministry of Justice, Chinese nationals are the largest immigrant group, accounting for 27.7% of foreign residents as of December 2019. The rate of Chinese women marrying Japanese men increased by 13.8% from 1995 to 2015 [6]. In general, Chinese women raising preschool children experience higher parenting stress compared to Japanese, Korean, and Brazilian women [7]; this higher parenting stress is also evident in immigrant Chinese women in Japan who are caring for their preschool-aged children [8]. Moreover, in contrast to Japanese women who believe that mothers should focus on taking care of their children and take on most of the responsibility of child-rearing, Chinese women believe that parents should share the responsibility of child-rearing and should obtain social support from their community and family, especially the woman’s mother and mother-in-law [9]. However, due to the different child-rearing values, immigrant Chinese women have difficulty obtaining child-rearing support from their mother-in-laws. Meanwhile, the child-rearing backing of the woman’s mother is also difficult to obtain because of living in a foreign country. Therefore, Chinese women are prone to experiencing stress related to both child-rearing and acculturation, given the lack of social support for child-rearing in Japan (which relates to maintaining traditional culture), and to adapting to new child-rearing values [10]. These factors and the lack of effective social support may affect their quality of life, including their mental and physical health.

Many studies have focused on postpartum immigrant women with poor mental health and a lack of postpartum mental health support [11,12]. Previous studies of postpartum immigrant Chinese women in Japan indicated that these new mothers experienced high stress and worse mental health, with low household income as a critical factor contributing to mental health [10,13]. However, severe mental health issues and high maternal stress are also found in women with preschool children [14,15]. Thus, it is necessary to investigate the health status of immigrant women with preschool children.

Moreover, a high level of parenting stress can give rise to serious health problems such as depression and sleep deprivation [16,17]. In a recent study on immigrant women, both parenting stress and acculturative stress were associated with social support [18]. Another study reported significant relationships between social support, acculturative stress, and mental health of immigrant women [19]. To improve immigrant women’s health, it is necessary to identify the relationships among social support, stress, and health. Nonetheless, few studies have examined these aspects in immigrant women in Japan. Thus, the question, “what are the associations of social support, parenting stress, acculturative stress, mental health, and physical health among immigrant Chinese women in Japan?” was considered in this study.

Norbeck and Tilden reported that social support directly impacts stress and health and has an indirect impact on health through stress [20]. A study on immigrant women showed that social support plays an indirect effect on physical and mental health through acculturative stress [21]. Another previous study indicated that social support from family has both direct and indirect effects on health through parenting stress among the Chinese population [22]. Therefore, another research question, “how did social support affect physical and mental health through parenting stress and acculturative stress among immigrant Chinese women in Japan?” was considered.

Norbeck presented a conceptual model regarding the mediating effect of stress between social support and health [23]. The mediating effects of acculturation stress and parenting stress among immigrant populations were found between mental health and social behavior, and between parents’ beliefs and mental health, respectively [24,25]. Hence, the question “do acculturative stress and parenting stress play a moderating role in the relationship between social support and health among immigrant Chinese women in Japan?” was considered.

Based on Norbeck’s conceptual model and the above-mentioned research findings, we constructed a theoretical framework of social support, stress, and health, as shown in Figure 1a. Stress was evaluated from parenting stress and acculturative stress, considering immigrant Chinese women in Japan mainly suffer stress from child-rearing and acculturation. Within the structural model, we had three hypotheses as follows.

**Hypothesis** **1.**
*Social support may directly affect parenting stress, acculturative stress, mental health, and physical health (see Figure 1b).*


**Hypothesis** **2.**
*Social support may indirectly affect mental health and/or physical health through acculturative stress and/or parenting stress (see Figure 1c).*


**Hypothesis** **3.**
*Parenting stress and/or acculturative stress may mediate the association between social support and mental health, as well as the association between social support and physical health (see Figure 1d).*


In order to test the theoretical model and hypotheses and to better understand the relationships among social support, parenting stress, acculturative stress, mental health, and physical health, structural equation modeling (SEM) was used to construct a structural model. SEM is a useful and powerful tool to evaluate a theoretical model of relationships among variables in a cross-sectional study and to measure the moderating effect of variables [26,27]. Further, to ensure the validity and reliability, the methodology of survey and SEM with realism paradigm were referenced [28].

This study aimed to (1) evaluate the demographic factors contributing to social support, parenting stress, acculturative stress, mental health, and physical health; (2) identify the relationships among social support, parenting stress, acculturative stress, mental health, and physical health; and (3) propose a structural model among social support, acculturative stress, parenting stress, mental health, and physical health in immigrant Chinese women in Japan.

## 2. Materials and Methods

### 2.1. Research Design, Participants, and Data Collection

This study used a descriptive cross-sectional design wherein quantitative data were collected and analyzed. Immigrant Chinese women were recruited from five regions in Japan between 1 March and 31 October 2019. Participants were recruited from various community settings, including childcare seminars for foreign parents, child-rearing seminars for Chinese mothers, nursery schools, and bureaus for the future of children. The snowball sampling method was used to recruit more participants. Inclusion criteria were as follows: Chinese women living in Japan; women with at least one child under the age of 6; women able to speak, read, and understand Chinese.

This study used an anonymous self-administered questionnaire. The questionnaire was in Chinese and consisted of an informed consent instruction; demographic information; and questions about social support, parenting stress, acculturative stress, and quality of life. The questionnaire was distributed using three methods.

First was distribution via community settings, where the participant was recruited. We explained the research content either in writing or verbally to the principal or the person in charge of the nursery school or children’s hall. The informed consent instruction was presented, and permission for research cooperation was obtained. Survey forms and request texts were distributed to the participants through the nursery school and bureaus for the future of children. Cooperation in the research was voluntary and we ensured that consent was obtained for what was returned. We provided participants with a reply envelope, inside which they placed their completed survey form and posted it to the mailbox. We then collected the envelope through the mail.

Second was the snowball sampling method. We were introduced to Chinese women through other Chinese acquaintances. The research contents were explained, and the questionnaire and request text were distributed to those who provided permission for research cooperation. Again, we provided participants with a reply envelope and collected their responses by mail.

Third was the face-to-face distribution method. We participated in childcare seminars for foreign parents and child-rearing seminars for Chinese mothers, where many foreign women, including Chinese women, gathered. The research contents were explained, and the questionnaire and request text were distributed to those who provided consent for research cooperation. The researchers directly collected responses after the questionnaire was completed. If it was difficult to collect it on the spot, we provided the participants with a reply envelope and collected it by mail.

### 2.2. Measures

#### 2.2.1. Social Support

Social support was examined based on the child-rearing values of Chinese women that would like to obtain social support from their community and family, using three domains including family, friends, and significant others, via the Multidimensional Scale of Perceived Social Support (MSPSS) developed by Zimet et al. [29]. The scale was translated into Chinese by Huang et al. [30] and the validity of the MSPSS Chinese version was previously estimated [31,32]. The instrument consists of 12 items; each item was scored on a seven-point Likert-type scale ranging from 1 (very strongly disagree) to 7 (very strongly agree). The scale scores were computed by summing the item scores; higher scores indicated a higher level of perceived social support. Cronbach’s alpha in this study sample was 0.946.

#### 2.2.2. Parenting Stress

Parenting stress was assessed using the Child-Rearing Stress Scale (CRSS) for foreign mothers in Japan, developed by Shimizu and Masuda [33]; You and Emori translated the CRSS into Chinese and evaluated the validity [8]. The CRSS had been used to evaluate parenting stress for various populations in different countries [7,34] and consists of 40 items (“I feel vague anxiety when I think about childcare”; “Because of rearing children I feel tired”; etc.), from 10 subscales (“anxiety with childrearing”; “feeling restraint with childrearing”, etc.), with each item scored from 1 (strongly disagree) to 4 (strongly agree). The summary scores were added together, with a higher score indicating a higher level of parenting stress. In this study sample, Cronbach’s alpha was 0.959.

#### 2.2.3. Acculturative Stress

Acculturative stress was measured using the Chinese version of the Acculturative Stress Scale (ASS), developed by Sandhu et al. [35]; Zhang et al. translated the ASS into Chinese [36], and the validity was previously estimated for various populations, including the Chinese migrant population [21,37,38,39]. The instrument includes 36 items (“Homesickness bothers me”; “I feel sad living in unfamiliar surroundings”, etc.), from seven subscales (“homesickness”; “stress due to change/culture shock”, etc.), with each item rated on a five-point scale, ranging from 1 (strongly disagree) to 5 (strongly agree). The composite score was the sum of the scores from the 40 items, with a higher composite score indicating a higher level of acculturative stress. In this study sample, Cronbach’s alpha was 0.963.

#### 2.2.4. Quality of Life: Mental and Physical Health

Mental and physical health were measured using the Chinese version of the SF-36 version 2 (SF-36v2) health survey, which was provided by QualityMetric, Inc. The Chinese SF-36v2 has previously been used to estimate the validity and reliability in general populations [40,41]. The SF-36v2 is a very general measure that evaluates quality of life, including mental health and physical health, and consists of 36 items from eight domain scales and measures two dimensions: mental health, presented by a mental component summary (MCS) including four domain scales (vitality, social functioning, role-emotional, and mental health); and physical health, presented by a physical component summary (PCS) including four domain scales (physical functioning, role-physical, bodily pain, and general health). The scores were calculated by the PRO CoRE software, which was provided by QualityMetric, Inc. The original scores of the MCS and PCS measures have been transformed to norm-based T scores ranging from 0 (worse health state) to 100 (best health state), having a mean of 50 and a standard deviation of 10. Thus, scores above and below 50 are above and below the average, respectively, in the general population. Group mean scores less than 47 indicated the presence of impaired functioning in the associated dimension [42]. In this study sample, Cronbach’s alpha of MCS and PCS were 0.825 and 0.706, respectively.

#### 2.2.5. Demographics

Based on the literature review [5], a brief demographic survey was developed for the study, which collected data on age, number of children, duration of residence, employment status, education level, status of residence, family structure, annual household income, Japanese proficiency, nationality of spouse, and employment status of spouse. There are 29 types of statuses for foreign residents, which were categorized by three domains including work permit, non-work permit, and family permit. Japanese proficiency was assessed using self-scoring by four items including speaking, reading, writing, and listening. Each item was rated on a four-point scale, ranging from 1 (cannot do it at all) to 4 (can do it exactly), with a higher score indicating a higher perceived Japanese proficiency.

### 2.3. Data Analysis

Using IBM SPSS Statistics version 26.0, the collected data were analyzed. Descriptive statistics were used to analyze participants’ demographic characteristics. The distribution of continuous variables was assessed using the Shapiro–Wilk normality test. The Mann–Whitney U test and the Kruskal–Wallis test were used to evaluate significant differences in the non-parametric data obtained from MSPSS, CRSS, ASS, PCS, and MCS based on sociodemographic factors. Spearman’s rank correlation coefficient was calculated to examine the bivariate relationships among MSPSS, CRSS, ASS, PCS, and MCS. To reduce the possibility of a type I error occurring from multiple comparisons, Bonferroni correction was used to adjust for the *p*-value [43]. AMOS Version 26.0 was used to implement SEM. The ratio of chi-square divided by the degrees of freedom (CMIN/DF), the standardized root mean square residual (SRMR), the root mean square error of approximation (RMSEA), the goodness-of-fit index (GFI), and the adjusted goodness-of-fit index (AGFI) were used to assess the model fit. RMSEA < 0.05, SRMR < 0.05, a small CMIN/DF, and GFI and AGFI close to 1.00 indicate a good fit [44]. The extension of the causal steps approach was used to test the mediating effect [27]. The results are presented as 95% confidence intervals.

### 2.4. Ethics Statement

This study was approved by the ethical review committee of Hokkaido University, Japan, and performed in accordance with the ethical standards as laid down in the 1964 Declaration of Helsinki and its latter amendments. Informed consent was obtained from each participant included in the study before they completed the study questionnaire.

## 3. Results

A total of 150 questionnaires were distributed, and 112 (recovery percentage of 74.7%) were completed and returned. Among the completed questionnaires, three did not respond to items representing major variables and were thus excluded. The final sample included 109 (97.3% of the total sample) participants for the analyses.

### 3.1. Relationships between the Demographic Factors and the Main Variables

Participants’ mean age was 33.9 (standard deviation (SD) = 5.6) years, and all were married. Their spouses’ average age was 36.2 (SD = 7.4) years. The participants’ demographic characteristics and the relationships between the demographic factors and the main variables are presented in Table 1. Most women (40.3%) had lived in Japan for more than 10 years and were employed (66.1%). The largest proportion of participants had a nuclear family structure (83.4%), a middle-level (30,000–70,000 USD) annual household income (51.4%), and a high level of Japanese proficiency (51.4%). The majority of spouses were Chinese (81.7%) and were employed (97.2%).

Women with one child showed a significantly lower mental health status than women with more than two children (*p* = 0.028). Shorter residence duration was associated with a higher level of acculturative stress (*p* = 0.013) and a lower level of mental health (*p* = 0.003). Employed women had significantly lower parenting stress (*p* = 0.032) and significantly better mental health (*p* = 0.042) than unemployed women. A middle-level (30,000–70,000 USD) annual household income was associated with weaker social support (*p* = 0.038), higher acculturative stress (*p* = 0.018), higher parenting stress (*p* = 0.016), and worse physical health (*p* = 0.003) than a high-level annual household income (>70,000 USD). A high level of Japanese proficiency was associated with significantly stronger social support (*p* = 0.012), lower acculturative stress (*p* = 0.003), and better mental health (*p* = 0.049). Women who had a Japanese spouse had significantly stronger social support than women who had a Chinese spouse (*p* = 0.043).

### 3.2. Bivariate Correlation Coefficients among Social Support, Parenting Stress, Acculturative Stress, Mental Health, and Physical Health

The median scores (interquartile range) for social support, acculturative stress, parenting stress, mental health, and physical health were 65.0 (16.0), 79.0 (31.0), 81.0 (27.0), 46.1 (13.1), and 52.5 (10.9), respectively. Table 2 shows the bivariate correlation coefficients among the main variables. Social support was negatively correlated with acculturative stress (*r* = −0.296, *p* = 0.002) and parenting stress (*r* = −0.324, *p* = 0.001) and positively correlated with mental health (*r* = 0.407, *p* = 0.000). Acculturative stress was positively correlated with parenting stress (*r* = 0.546, *p* = 0.000) and negatively correlated with mental health (r = −0.360, *p* = 0.000) and physical health (*r* = −0.299, *p* = 0.002). Parenting stress was negatively correlated with mental health (*r* = −0.554, *p* = 0.000) and physical health (*r* = −0.334, *p* = 0.000).

### 3.3. Results of the Structural Model Analysis

Figure 2 shows the results of the structural model analysis. The final path model of the relationships among the main variables and the process of testing the mediating effects of parenting stress and acculturative stress between social support and health are presented.

#### 3.3.1. Path Model of the Relationships among the Main Variables

SEM was used to test the hypothetical path model. Nonsignificant direct paths were removed one by one, and then a best good fit model was proposed. The best good fit structural model of the relationship among social support, parenting stress, acculturative stress, mental health, and physical health is shown in Figure 2b. Since the path coefficients from social support to physical health did not show an acceptable good fit model, this path was removed from the final model [45]. The summary of standardized effect results for the path model is presented in Table 3.

The final path model of the relationships among social support, acculturative stress, parenting stress, mental health, and physical health showed a good fit (*x^2^* = 0.009, *p* = 0.926, *df* = 1, CMIN/DF = 0.009, RMSEA = 0.000, GFI = 1.000, AGFI = 1.000, SRMR = 0.002). In the path model, social support showed a significant direct effect on parenting stress (*β* = −0.21, *p* < 0.05), acculturative stress (*β* = −0.19, *p* < 0.05), and mental health (*β* = 0.31, *p* < 0.001) but not on physical health. Social support showed a significant indirect effect on physical health (*β* = 0.09, *p* < 0.05). Parenting stress had a significant direct effect on mental health (*β* = −0.38, *p* < 0.001) and physical health (*β* = −0.24, *p* < 0.05). Acculturative stress did not show a significant direct effect on either mental health (*β* = −0.08) or on physical health (*β* = −0.19).

#### 3.3.2. The Mediating Effect of Stress between Social Support and Health

To establish the mediation, the independent variable must be shown to affect the mediator and dependent variables [27]. Since acculturative stress did not show a significant effect on mental health and physical health, only parenting stress as a mediating variable was tested, and the variable of acculturative stress was ignored. Using the bootstrapping method, two steps were performed. In Step 1, the relationships between the independent variable (social support) and dependent variables (mental health and physical health) were evaluated (Figure 2a). Results showed that a significant direct effect from social support to mental health was evaluated (*β* = 0.40, *p* < 0.001; Figure 2a) but a nonsignificant direct effect from social support to physical health was found (*β* = 0.08; Figure 2a). Therefore, in Step 2, the mediating effect of social support and physical health was ignored and rather, the mediating effect of parenting stress between social support and mental health was evaluated. Results suggest that parenting stress significantly partially mediated the association of social support and mental health ((*β* = 0.09, *p* < 0.05, Figure 2b).

## 4. Discussion

The immigrant Chinese women in this study experienced a high level of parenting stress when compared with previous studies on various populations, such as Japanese mothers, Korean mothers, and immigrant Brazilian women in Japan, which used a similar child-rearing scale [7,34]. Moreover, this study sample showed a lower level of mental health and a higher level of social support than reported in a previous study of immigrant women [21]. Despite greater perceived social support, Chinese women in Japan may experience more mental stress related to living in a strange country. However, immigrant Chinese women in Japan experienced a lower level of acculturative stress than immigrant Chinese residents in Australia and the United States [37,46]. This may be because China and Japan share an Asian cultural background and thus have a smaller cultural gap than that with Western cultures.

This study showed that immigrant Chinese women’s residence duration in Japan was significantly associated with acculturative stress and mental health. Acculturation is an unfolding process, and time spent in the destination culture plays a prominent role [47]. This suggests that it is necessary to improve mental health among newcomer immigrant Chinese women by helping them adapt to Japanese culture. Employed women in this study experienced lower parenting stress and better mental health than those who were unemployed. This is consistent with previous studies’ findings [16,48]. In China, the rate of employment among 18- to 64-year-old women is reported to be 71.7% [49], suggesting that caring for children while working is a universal trait among Chinese women. Thus, we may suggest the Japanese government provide more employment opportunities for immigrant women in Japan to reduce their parenting stress and enhance their mental health.

Annual household income had a significant association with social support, acculturative stress, parenting stress, and physical health, especially among women with a middle-level annual household income. Many studies have focused on low-income immigrant women [50,51], but middle-income immigrant women have generally been ignored. This is the same in Japan, where low-income expectant mothers have specifically been identified as at risk, socially speaking [52]. Our findings suggest that social support providers and public health nurses should not ignore middle-income immigrant women, despite their relatively stable finances. Additionally, Japanese proficiency was associated with social support, acculturative stress, and mental health. Language barriers were recognized as especially difficult to handle among immigrant women, strongly affecting their health [53,54]. Furthermore, the spouse’s nationality was associated with social support. This might be because a Japanese husband will know more about the Japanese social system and policies, which can help immigrant Chinese women obtain more social support.

We found a significant association between acculturative stress and parenting stress, and both acculturative stress and parenting stress were associated with social support among immigrant Chinese women. In this study, acculturative stress showed the most association with parenting stress. A previous study among immigrant women reported similar correlations among social support, parenting stress, and acculturative stress, but acculturative stress and social support showed most correlation [18]. This study could identify that the relationship between acculturative stress and parenting stress is synchronous and interactive. Developing positive and culturally appropriate social support for immigrant mothers, decreasing parenting stress, and increasing cultural adaptation to the host country’s culture are essential requirements that cannot be ignored by social support providers and public health nurses.

Additionally, mental health and physical health showed a negative but nonsignificant relationship in this study. A possible explanation is that women with poor physical health had more regular social activities such as employment, which may have increased their sense of security, thereby improving their mental health [55]. Meanwhile, the results suggest that acculturative stress for Chinese immigrant mothers did not mediate the relationship between social support and health. These findings are inconsistent with those of a previous study of immigrant women in South Korea where acculturative stress directly impacts mental health and social support indirectly impacts mental health through acculturation [21]. This may suggest that cultural differences between the host and native countries could influence the relationships among social support, acculturative stress, and mental health in immigrant women. Therefore, further research is needed to explore the role of acculturative stress between social support and health among the female immigrant population.

In the path model, the results confirmed Hypothesis 1, that social support directly affects parenting stress, acculturative stress, and mental health but not physical health. The reason there was no direct effect of social support on physical health might be that this study’s sample had an above-average physical health level [42]. Moreover, Hypothesis 2, which states that social support indirectly affects mental health through parenting stress, was partially confirmed. This result is consistent with those of a previous study among mothers with disabled children [56]. Furthermore, Hypothesis 3, which states that parenting stress as a mediating variable partially mediated the association between social support and mental health, was also partially confirmed. Additionally, the path relationship among social support, parenting stress, and mental health confirmed Norbeck’s model of the relationship among social support, stress, and health.

Based on the structural model and these confirmed hypotheses, there are three suggestions about the mental health promotion of immigrant women for social support providers and public health nurses. First, social support could be directly provided to improve mental health. This suggestion has been generally implemented for immigrant populations in many countries including Japan [10,57,58]. Second, mental health support providers need to focus on stress not only from the acculturation process but also from the child-rearing process among immigrant women. Some mental health promotion programs and interventions only focused on cross-cultural stress while ignoring parenting stress among immigrant women [12,59]. Third, since parenting stress could influence the effectiveness of mental health support, more attention needs to focus on parenting stress to ensure the effectiveness of mental health support to avoid wasting social resources. A practical and evidence-based mental health promotion program for immigrant Chinese women should be developed in the future. Moreover, many studies examined the moderating effect in the relationships among social support, stress, and health [19,60]. Future studies should further evaluate the intrinsic relationships among social support, acculturative stress, parenting stress, mental health, and physical health in immigrant women, and the mediating effect as well as the moderating effect.

This study has several limitations that must be considered. First, the sample size was small, which may affect the generalizability of the results; there may be differences between different regions in Japan, and the study sample did not include all regions. Moreover, women with older children could experience different parenting stress than those with younger children; this study did not subdivide children by age, which might have influenced the results. Furthermore, stress is not a single variable but consists of many factors and processes. The cross-sectional design is also a limitation; thus, future studies should employ a longitudinal design to further explore the relationships examined in this study.

This study proposed a framework to show the relationship among social support, stress, and health, which fills a gap in this research field for immigrant Chinese women and immigrant women in Japan. Immigrant Chinese women are a rapidly growing group in many countries and the largest in Japan. We look forward to the application of the framework among immigrant women in future studies. Moreover, a partially mediating effect of parenting stress between social support and mental health was identified, which provides evidence for social support providing services and public health services to ensure the effectiveness of social support on improving mental health. Since the different acculturative background may influence the results of the psychometric test, it is important to access psychometric tests for miscellaneous immigrant women providing scientific evidence for improving their health.

## 5. Conclusions

Immigrant Chinese women in Japan demonstrated high levels of parenting stress and low levels of mental health. The number of children, duration of residence, employment status, annual household income, Japanese proficiency, and nationality of spouse were examined as significant factors associated with social support, parenting stress, acculturative stress, mental health, and physical health. Moderate associations were found between social support and mental health and between acculturative stress and parenting stress. The path model showed that social support directly influenced parenting stress, acculturative stress, and mental health but not physical health. In addition, the mediating effect of parenting stress between social support and mental health was evaluated. A broader useful framework among immigrant women was confirmed, which reinforces the importance of reducing parenting stress to maximize social support’s protective effect on mental health. A practical and evidence-based mental health promotion program for immigrant Chinese women in Japan should be developed in a future study.

## Figures and Tables

**Figure 1 healthcare-09-00258-f001:**
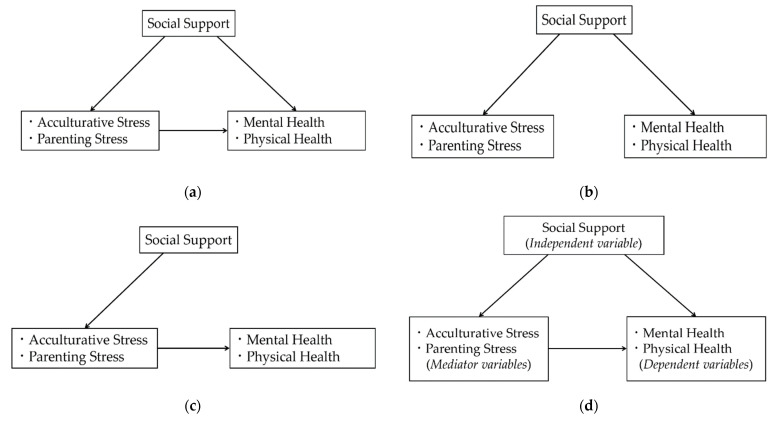
(**a**) Theoretical framework of the relationships among social support, acculturative stress, parenting stress, mental health, and physical health; (**b**) Hypothesis 1: direct effect among main variables; (**c**) Hypothesis 2: indirect effect among main variables; (**d**) Hypothesis 3: the mediating effect of stress between social support and health.

**Figure 2 healthcare-09-00258-f002:**
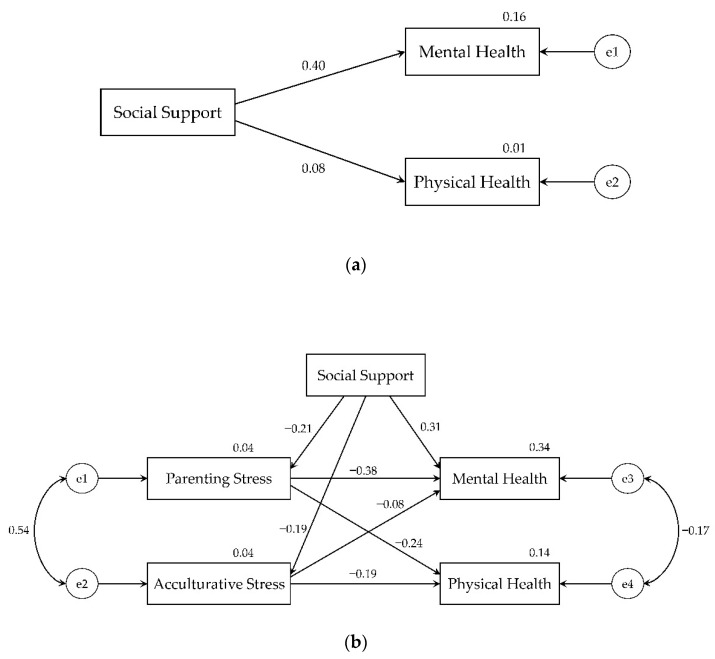
The results of structural model analysis (*n* = 109; e1−e4: error terms). (**a**) Step 1 of testing the mediating effect; (**b**) The final path model and step 2 of testing the mediating effect.

**Table 1 healthcare-09-00258-t001:** The relationships between the demographic factors and the main variables (*n* = 109).

		MSPSS ^1^		ASS ^2^		CRSS ^3^		MCS ^4^		PCS ^5^		
Variables	*n* (%)	M ^6^ (IQR ^7^)	*p*-Value	M(IQR)	*p*-Value	M(IQR)	*p*-Value	M(IQR)	*p*-Value	M(IQR)	*p*-Value	
Age (years)											
20–29	19 (17.4)	65.0 (12.0)	0.447	83.0 (36.0)	0.504	83.0 (23.0)	0.534	45.6 (32.3)	0.183	52.2 (11.7)	0.394
30–39	78 (71.6)	63.5 (18.0)		77.5 (30.0)		81.5 (29.0)		44.4 (14.0)		52.6 (10.6)	
40–49	10 (9.2)	71.0 (10.0)		74.0 (39.0)		77.0 (34.0)		51.8 (11.2)		50.3 (10.9)	
>50	2 (1.8)	58.0		91.5		95.5		47.7		58.7	
Number of children											
1	85 (78.0)	65.0 (16.0)	0.703	79.0 (30.0)	0.959	82.0 (23.0)	0.229	44.2 (11.8)	0.028	52.2 (11.8)	0.852
≥2	24 (22.0)	62.5 (18.0)		78.0 (32.0)		71.5 (35.0)		48.5 (16.3)		52.6 (7.5)	
Duration of residence (years)											
<5	33 (30.3)	64.0 (19.0)	0.378	85.0 (42.0)	0.013	84.0 (24.0)	0.136	43.6 (11.7)	0.003	53.3 (13.1)	0.640
5–10	32 (29.4)	64.0 (18.0)		80.5 (23.0)		80.5 (23.0)		42.9 (11.2)		50.2 (8.5)	
>10	44 (40.3)	71.0 (14.0)		72.0 (25.0)		78.0 (30.0)		49.4 (13.2)		52.9 (9.6)	
Employment status											
Employed	72 (66.1)	65.5 (13.0)	0.201	79.0 (32.0)	0.687	78.0 (27.0)	0.032	47.5 (11.4)	0.042	51.9 (11.1)	0.321
Unemployed	37 (33.9)	63.0 (24.0)		77.0 (32.0)		85.0 (23.0)		42.9 (12.4)		53.1 (11.4)	
Education level											
High school	32 (29.4)	65.0 (17.0)	0.409	80.0 (43.0)	0.117	80.5 (30.0)	0.895	47.1 (11.0)	0.566	51.3 (9.5)	0.470
University	41 (37.6)	63.0 (18.0)		81.0 (29.0)		79.0 (26.0)		44.2 (13.1)		52.5 (13.2)	
Graduate school	36 (33.0)	71.0 (15.0)		72.0 (29.0)		82.0 (23.0)		45.6 (17.8)		52.9 (9.6)	
Status of residence											
Work permit	24 (22.0)	69.5 (13.0)	0.534	79.0 (22.0)	0.053	78.5 (18.0)	0.187	46.1 (15.1)	0.169	54.1 (10.9)	0.127
Non-work permit	37 (33.9)	64.0 (23.0)		86.0 (34.0)		88.0 (23.0)		42.7 (9.4)		50.7 (12.3)	
Family permit	43 (39.5)	66.0 (13.0)		75.0 (32.0)		81.0 (31.0)		48.1 (14.1)		52.2 (10.4)	
No answer	5 (4.6)										
Family structure											
Nuclear family	91 (83.4)	65.0 (17.0)	0.736	79.0 (30.0)	0.874	82.0 (28.0)	0.793	48.1 (13.9)	0.392	52.6 (9.6)	0.167
Extended family	15 (13.8)	63.0 (14.0)		72.0 (38.0)		85.0 (25.0)		44.5 (13.1)		45.5 (12.5)	
No answer	3 (2.8)										
Annual household income (USD)											
<30,000	30 (27.5)	64.0 (20.0)	0.038	79.0 (36.0)	0.018	78.5 (35.0))	0.016	44.0 (9.3)	0.218	52.8 (9.8)	0.003
30,000–70,000	56 (51.4)	63.0 (17.0)		80.5 (26.0)		85.0 (23.0)		45.5 (14.5)		49.7 (10.8)	
>70,000	22 (20.2)	71.0 (14.0)		66.0 (31.0)		72.0 (20.0)		49.6 (10.6)		55.7 (6.4)	
No answer	1 (0.9)										
Japanese proficiency											
High	56 (51.4)	71.0 (13.0)	0.012	72.0 (27.0)	0.003	80.5 (34.0)	0.067	48.1 (14.8)	0.049	51.9 (11.9)	0.985
Low	53 (48.6)	62.0 (20.0)		82.0 (30.0)		83.0 (24.0)		42.9 (10.1)		52.8 (9.5)	
Nationality of spouse											
Chinese	89 (81.7)	64.0 (20.0)	0.043	80.0 (30.0)	0.281	83.0 (24.0)	0.066	44.2 (12.5)	0.055	52.2 (10.9)	0.675
Japanese	19 (17.4)	72.0 (14.0)		72.0 (32.0)		72.0 (32.0)		47.8 (13.6)		53.1 (12.1)	
No answer	1 (0.9)										
Employment status of spouse											
Employed	106 (97.2)	65.0 (15.0)	0.591	79.0 (31.0)	0.078	81.0 (27.0)	0.133	46.4 (13.6)	0.036	52.3 (11.0)	0.531
Unemployed	3 (2.8)	56.0		118.0		106.0		35.1		53.8	

^1^ MSPSS: Multidimensional Scale of Perceived Social Support; ^2^ ASS: Acculturative Stress Scale; ^3^ CRSS: Child-Rearing Stress Scale; ^4^ MCS: Mental Component Summary; ^5^ PCS: Physical Component Summary; ^6^ M: median; ^7^ IQR: interquartile range.

**Table 2 healthcare-09-00258-t002:** Bivariate correlation coefficients among the main variables (*n* = 109).

Variables	MSPSS ^1^		ASS ^2^		CRSS ^3^		MCS ^4^		PCS ^5^	
	Correlation Coefficient	*p*-Value ^6^	Correlation Coefficient	*p*-Value	Correlation Coefficient	*p*-Value	Correlation Coefficient	*p*-Value	Correlation Coefficient	*p*-Value
MSPSS	1.000									
ASS	−0.296	0.002	1.000							
CRSS	−0.324	0.001	0.546	0.000	1.000					
MCS	0.407	0.000	−0.360	0.000	−0.554	0.000	1.000			
PCS	0.166	0.084	−0.299	0.002	−0.334	0.000	0.095	0.324	1.000	

^1^ MSPSS: Multidimensional Scale of Perceived Social Support; ^2^ ASS: Acculturative Stress Scale; ^3^ CRSS: Child-Rearing Stress Scale; ^4^ MCS: Mental Component Summary; ^5^ PCS: Physical Component Summary. ^6^
*p*-Value ≤0.005 considered statistically significant after Bonferroni correction.

**Table 3 healthcare-09-00258-t003:** Summary of standardized effect results for the structural model (*n* = 109).

Standardized Effect Results between Variables	Direct Effect	Indirect Effect	Mediation Result
Social support → Parenting stress	−0.21 *		
Social support → Acculturative stress	−0.19 *		
Parenting stress → Mental health	−0.38 ***		
Parenting stress → Physical health	−0.24 *		
Acculturative stress →Mental health	−0.08		
Acculturative stress → Physical health	−0.19		
Social support → Parenting stress → Physical health	ns	0.09 *	None
Social support → Parenting stress → Mental health	0.31 ***	0.09 *	Partial

*** *p* < 0.001; * *p* < 0.05; ns: nonsignificant.

## Data Availability

Not applicable.

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
