# Peer review of "Relationships of Social Support, Stress, and Health among Immigrant Chinese Women in Japan: A Cross-Sectional Study Using Structural Equation Modeling"

_healthcare, 2021, doi:10.3390/healthcare9030258_

Round 1

Reviewer 1 Report

The paper submitted presents a high structural quality in terms of writing, rationale, methods, and design. I also believe that it meets the requirements in its current version for publication, also referring to careful and rigorous analysis.

However, the paper reports data from a very concrete population without explaining convincingly how this data might be of interest to a more general audience. Without a clear explanation of this point, as well as an introduction explaining this with literature, it is not clear that this paper is interesting to be published in this journal.

Therefore, I even consider that the paper should be re-oriented in its approach to a more general audience.

Author Response

Response to Reviewer 1 Comments

The paper submitted presents a high structural quality in terms of writing, rationale, methods, and design. I also believe that it meets the requirements in its current version for publication, also referring to careful and rigorous analysis. However, the paper reports data from a very concrete population without explaining convincingly how this data might be of interest to a more general audience. Without a clear explanation of this point, as well as an introduction explaining this with literature, it is not clear that this paper is interesting to be published in this journal. Therefore, I even consider that the paper should be re-oriented in its approach to a more general audience.

Response: We gratefully thank you for the time spent giving us suggestions and comments. As you have pointed out, since the data of this study from a concrete population, the reason should be explained. We have added the explanation about the necessities of focusing on this concrete population in the introduction (page 2, line 14 to 20).
Moreover, we have added a paragraph to discuss the wider implications of the study findings and specify future research avenues in the last paragraph of the discussion (page 11, line 38 to 48). Furthermore, since the structural model is the most important result in this study, we have added the structural equation modeling approach in the title to attract a more general audience. Besides, according to the suggestions and comments from another reviewer, we have revised the introduction and discussion more systematically. We have also added research questions to the introduction (page 2, line 21 to 45). Thank you again for your suggestions and comments. If we have misunderstood your comments, please give us a chance to revise the manuscript again.

Reviewer 2 Report

For the authors’ guidance my evaluation and some constructive remarks that would help to improve the paper’s quality are included below:

(1) First of all, I congratulate the authors for this excellent scientific paper. I think the paper has a strong and original argument. The research argument is streamlined in a way that addresses the core issues in a systematic and effective manner. The authors have successfully linked up some perspectives and concepts of various disciplines. The conceptualisation has enriched a substantial depth and theoretical model.

(2) The study focuses on describing the method from a textbook perspective instead of actually explaining how the method was carried out. Hence, it is a bit difficult to ascertain the trustworthiness of the research.

(3) Upfront, the authors ought to ensure a roadmap for the readers indicating how the argument, research design (e.g., research methodology, research paradigm, etc.), and theoretical assertions are interconnected, systematically. They must clarify the ontological, epistemological, hermeneutic, and phenomenological presumptions of the research paradigm and methodology used in a paradigmatic manner.

(4) I recommend the authors to read “Marilyn Healy, Chad Perry, (2000) "Comprehensive criteria to judge validity and reliability of qualitative research within the realism paradigm", Qualitative Market Research: An International Journal, Vol. 3 Iss: 3, pp.118 – 126.” On page 121, Healy & Perry indicate the range of methodologies and their related paradigms dealing with the complexity and validity of their arguments. Moreover, the authors may enhance the methodology of the research by including some scientific works of E.G. Guba, Y.S. Lincoln, N.K. Denzin and so forth.

(5) I may recommend the authors an engaged scholarship approach that can synthesise the viewpoints and attitudes of “academia”, “government”, and “practitioners” in terms of theory-practice combination. How social support approach influences public policy discourse? Can it be successful for clarifying dialectical relations in the framework of the research limitation?

(6) I think the authors may imply the direct effect of stakeholder theory (i.e., both key stakeholders and miscellaneous stakeholders) on public policy. Perhaps, the role of key stakeholders – both “public” stakeholders and “civil society” stakeholders – in the development of can be worth further investigation.

(7) I believe the authors will be more successful if they focus on clear, direct arguments that build up one at a time and add up sequentially to persuasive and tightly focused cases.  My own struggles with the task of writing peer-reviewed articles suggest that it is worth spending more time in the detailed planning of papers around a sequence of bullet-point arguments. Plans are best if argument-driven. It is worth resisting writing full prose until a very detailed plan has been developed that seems to work, and then supporting material can be added after each bullet point argument.

(8) The paper lacks a clear (and interesting) research question, and this could also be linked to the fact that a theoretical part is seemingly a bit weak. The readers understand somehow “what” the paper attempts to do, they also get the “how”, however the most interesting question, the “why!!!” is not tackled.

“What-Context”: It is about the descriptive nature (e.g., quotations of the relevant literature, clarifying notions, etc.) of this study.

“How-Context”: It is associated with the “methodology and research tools (e.g., argumentation, justification, research enquiry and so on.)” used in this investigation.

“Why-Context”: It is more relevant to the originality, added value, specific contribution to the existing literature, implications for future studies, and so on. I recommend the authors to enrich the “Why-Context.” The authors should give more detailed clarifications about the research findings and the originality of this investigation.

(9) It should be demonstrated precisely how the authors come up with the research questions, and why it is interesting and relevant to the field. Linked to that, I am wondering about the (theoretical) contribution of the paper. To me, it will be more precise to make a direct connection with the theoretical part and research outcomes through stating research enquiries.

(10) Consequently, I appreciate there has been a lot of reading and ground covered. However, the study should have a stronger focus, compelling argument and discussion, and an indication of why the paper holds value to the readership of MDPI – Healthcare. All in all, I recommend the authors to reconsider the approach adopted here; think about the main empirical question they wish to examine; make sure the literature review is a lot more cohesive, and make sure the link between the research questions, research results, and the conclusion is a lot “tighter” than presented herewith.

(11) MDPI – Healthcare is a remarkable peer-reviewed journal. A referee ought to recommend a manuscript that is deemed a great contribution to the journal’s future achievements. Consequently, the paper demonstrates rigorous research outcomes/findings that can be useful for the readers of the MDPI – Healthcare.
